# Implementation of the Community Assets Supporting Transitions (CAST) transitional care intervention for older adults with multimorbidity and depressive symptoms: A qualitative descriptive study

Carrie McAiney[1][☯]*, Maureen Markle-Reid[2][☯], Rebecca Ganann[2][☯], Carly Whitmore[2][☯], Ruta Valaitis[2][‡], Diana J. Urajnik[3][‡], Kathryn Fisher[2][‡], Jenny Ploeg[2][‡], Penelope Petrie[2][‡], Fran McMillan[3][‡], Janet E. McElhaney[4][‡]

1 School of Public Health Sciences, University of Waterloo and Schlegel-UW Research Institute for Aging, Waterloo, Ontario, Canada, 2 Aging, Community and Health Research Unit, School of Nursing, McMaster University, Hamilton, Ontario, Canada, 3 Centre for Rural and Northern Health Research, Laurentian University, Sudbury, Ontario, Canada, 4 Northern Ontario School of Medicine and Health Sciences North Research Institute, Sudbury, Ontario, Canada

☯ These authors contributed equally to this work.
‡ RV, DJU, KF, JP, PP, FM and JEM also contributed equally to this work.
* carrie.mcainey@uwaterloo.ca

**Data Availability Statement:** In accordance with the Hamilton Integrated Research Ethics Board

## Abstract

### Background

Older adults with multimorbidity experience frequent care transitions, particularly from hospital to home, which are often poorly coordinated and fragmented. We conducted a pragmatic randomized controlled trial to test the implementation and effectiveness of Community Assets Supporting Transitions (CAST), an evidence-informed nurse-led intervention to support older adults with multimorbidity and depressive symptoms with the aim of improving health outcomes and enhancing transitions from hospital to home. This trial was conducted in three sites, representing suburban/rural and urban communities, within two health regions in Ontario, Canada.

### Purpose

This paper reports on facilitators and barriers to implementing CAST.

### Methods

Data collection and analysis were guided by the Consolidated Framework for Implementation Research framework. Data were collected through study documents and individual and group interviews conducted with Care Transition Coordinators and members from local Community Advisory Boards. Study documents included minutes of meetings with research team members, study partners, Community Advisory Boards, and Care Transition Coordinators. Data were analyzed using content analysis.

(HiREB) for Hamilton Health Sciences and McMaster University's Faculty of Health Sciences, data from this study cannot be shared publicly because of ethical restrictions involving potentially identifying information. The participant consent forms used in this study do not address open public access to the data. Data are available upon request from McMaster University, Faculty of Health Sciences, School of Nursing for researchers who meet the criteria for access to confidential data pending approval from the HiREB. For inquiries, please contact: Dr. Michael McGillion, Associate Professor and Assistant Dean, Research, School of Nursing, Faculty of Health Sciences, McMaster University, Email: mmcgill@mcmaster.ca; Phone: 905-525-9140 x 20275.

**Funding:** CM and MMR received funding from the Ontario SPOR Support Unit IMPACT award (grant no: 60502), https://ossu.ca/for-researchers/impact-awards/. RV, RG, MMR and CM received funding from the Labarge Foundation, McMaster University, http://optimalaging.mcmaster.ca/. This research was also undertaken, in part, thanks to the funding from Dr. Markle-Reid's Tier 2 CIHR Canada Research Chair, https://chru.mcmaster.ca/. The funders had no role in study design, data collection and analysis, decision to publish, or preparation of the manuscript.

**Competing interests:** The authors have declared that no competing interests exist.

## Findings

Intervention implementation was facilitated by: (a) engaging the community to gain buy-in and adapt CAST to the local community contest; (b) planning, training, and research meetings; (c) facilitating engagement, building relationships, and collaborating with local partners; (d) ensuring availability of support and resources for Care Transition Coordinators; and (e) tailoring of the intervention to individual client (i.e., older adult) needs and preferences. Implementation barriers included: (a) difficulties recruiting and retaining intervention staff; (b) difficulties engaging older adults in the intervention; (c) balancing tailoring the intervention with delivering the core intervention components; and (c) Care Transition Coordinators' challenges in engaging providers within clients' circles of care.

## Conclusion

This research enhances our understanding of the importance of considering intervention characteristics, the context within which the intervention is being implemented, and the processes required for implementing transitional care intervention for complex older adults.

## Introduction

A growing proportion of older adults ($\geq$ 65 years) live with two or more chronic conditions, referred to as multimorbidity [1]. Multimorbidity is common among older adults, experienced by approximately half of those over age 65 [2]. Population aging and rising life expectancies are key factors that contribute to this trend [3]. Of note, is that older adults with multimorbidity are twice as likely than those without it to also have depressive symptoms [4]. Depressive symptoms are serious and common among older adults yet often unrecognized and undertreated in this population [5]. Suboptimal treatment of depressive symptoms is associated with impaired function, more frequent use of health services [6], and reduced quality of life [5].

Older adults with multimorbidity are often high users of health care services, at higher risk of adverse events, and report poorer quality of life than those living with a single chronic condition [7–9]. For older adults with multimorbidity, navigating the transition from hospital to home is associated with substantial health-related challenges and fragmented care delivery [7]. Care transitions present numerous challenges in terms of integration and coordination of care across health sectors and primary/specialty providers involved in an individuals' circle of care. The consequences of fragmentated care include increased readmission rates, and decreased safety (e.g., medication errors) and patient satisfaction [9–11]. To address the challenges associated with poor transitions in care for older adults, interventions to enhance coordination, increase continuity of care, and improve the transitional care experience have been recommended [12–17]. Previous studies of care transitions have demonstrated reductions in hospital readmissions and other impacts, yet older adults with multimorbidity and those with mental health issues including depression, are often excluded from this research, leaving a gap in understanding of the impact of care transition interventions on these populations [18–20].

It is essential that interventions to address such gaps are evaluated to assess their impact. However, it is equally important to examine how such interventions are implemented in order to learn what is required to successful spread the interventions to other jurisdictions [21]. On average, evidence-based practices take 17 years to be adopted in routine practice [22–24]. Given the frequency of hospital-to-home transitions among older adults and the risk of

adverse events associated with poor transitions [7–11], examining the implementation barriers and facilitators can inform efforts to spread successful interventions [21, 25].

The purpose of this study is to describe the facilitators and barriers to implementing a new transitional care intervention for older adults with multimorbidity and depressive symptoms in three Canadian communities from the perspectives of the interventionists and community members.

## Methods

This study was part of a pragmatic randomized controlled trial (RCT) conducted to evaluate a new care transition intervention (Community Assets Supporting Transitions or CAST; ClinicalTrials.gov: NCT03157999). The specific design was a Type II hybrid effectiveness implementation study where implementation and effectiveness are given equal weight [26, 27]. This trial was conducted from 2017 to 2019.

To provide context, a description of the CAST intervention is provided as well as an overview of the study. Following this, the methods associated with the implementation study are provided. A more detailed description of the study, and findings from the effectiveness trial, are provided elsewhere [28, 29].

### The Community Assets Supporting Transitions (CAST) intervention

The CAST intervention was developed for older adults with multimorbidity and depressive symptoms. Building on a previous feasibility study [12], CAST was tested in 3 Ontario communities in Canada with the aim to improve the quality and experience of hospital to home transitions for older adults with multimorbidity and depressive symptoms.

The CAST intervention [28] consisted of an individualized patient- and caregiver-centred intervention for older adults with multimorbidity and depressive symptoms delivered by a community-based Registered Nurse acting as a Care Transition Coordinator (CTC) in addition to usual care. The CTCs participated in a standardized training program developed by the research team regarding intervention implementation. Clients of the CAST intervention (i.e., older adults with multimorbidity and depressive symptoms who were transitioning from hospital to home) received at least two home visits and a minimum of four telephone calls over the six-month intervention period. As part of the intervention, CTCs: (a) identified the health and social care professionals (e.g., family physician, specialists, home care providers, allied health professionals, community-based service providers) involved in the participant's circle of care and initiated a coordinated care plan with these individuals and organizations, including sending alerts to the participant's primary care provider regarding the presence of depressive symptoms, dementia, delirium, suicidal ideation, or medication issues; (b) identified and managed the patient's risk factors for depressive symptoms and other chronic conditions in accordance with evidence-based guidelines [30, 31]; (c) provided system navigation support, coordination, and follow up for participants and caregivers [32]; (d) conducted medication reviews and management in collaboration with participants, primary care providers and pharmacists using evidence-based best practice guidelines [30, 31]; (e) conducted problem-solving therapy with participants and caregivers using an established guide [33]; (f) implemented social and behavioural activation by assisting and encouraging participants to participate in regular physical activity programs tailored to social and behavioural needs; and (g) provided education to patients and their caregiver (e.g., regarding their chronic conditions and self-management strategies). CAST was designed to complement other supports and services, including home care. As a result, some older adults received services from both home care and the CAST CTCs. As this was part of a pragmatic trial, the CAST intervention could be tailored

to the needs and preferences of the clients receiving the CAST intervention. Clients could decide which elements, and how much of the intervention, they wanted.

## Overview of the pragmatic trial

Study participants were recruited within local community hospitals by trained recruiters. The recruiters used a standardized script to approach potentially eligible patients and determine their interest in learning about the study. When patients agreed to be screened for study eligibility, the recruiter completed an eligibility questionnaire, after which individuals were informed if they were eligible to participate. To be eligible, individuals had to: (a) be aged 65 years or older; (b) be planned for discharge from hospital to the community; (c) have at least two self-reported chronic conditions; (d) have depressive symptoms as measured by the two-item Patient Health Questionnaire (PHQ-2) [34]; (e) reside in one of the three participating regions with no plans to move out of the region during the trial; (f) be competent to provide informed consent, as determined by a score of $\geq$ 5 on the Short Portable Mental Status Questionnaire (SPMSQ) [35] or have a substitute decision-maker who could provide consent on their behalf; and (g) be competent in English (or French in one study site) or have an available interpreter.

This trial was conducted in three sites (Hamilton, Burlington, Sudbury) within the catchment area of two regional health authorities (Hamilton Niagara Haldimand Brant and North East Local Health Integration Networks) in Ontario, Canada. These communities were diverse as related to geography (two sites were suburban/rural, one site urban), socioeconomic, language, and ethno-cultural characteristics. Each site had one full-time equivalent CTC administering the intervention. In one site, the CTC role was split between two part-time Registered Nurses.

This trial aimed to recruit 216 participants across all three sites, however enrollment was below target (N = 127) [28]. More details about this clinical trial and the communities in which it was conducted, as well as the findings from the effectiveness study, are provided elsewhere [28].

## Collective impact

Given the complex nature of transitional care and the need for various partners and sectors to work together for hospital to home transitions to be successful, the Collaborative Intervention Planning Framework by Cabassa et al. [36] was used to enhance the potential for collective impact [37]. Collective impact involves a variety of key stakeholders coming together to address a complex issue by: establishing a shared vision of the issue; developing an accepted intervention to respond to the issue; agreeing on how the intervention will be evaluated; and establishing mechanisms for ongoing communication [37].

Prior to the initiation of the trial, the research team worked with key stakeholders in each community to organize and host a Community Forum. The Community Forums aimed to better understand the needs of each community in terms of transitional care, learn about existing transitional care initiatives, and gain insight into how the CAST intervention should be adapted to each community context. The research team worked collaboratively with the community stakeholders to determine who to invite to each community forum to reflect diverse perspectives. Attendees included older adults, family/friend caregivers, and representatives from hospitals, home care organizations, community support services, municipal governments, regional health care authorities, and other local organizations involved in care transitions.

Information obtained from the Community Forums was used to establish Community Advisory Boards (CABs) in each participating community. CAB members included: older adults, family/friend caregivers and members of the public (referred to as patient and public research partners); representatives from hospitals, home care organizations and community support services involved in hospital to home transitions; and members of the research team. The role of the CABs was to guide the local implementation of CAST in each community. This involved determining how CAST may need to be adapted to suit local conditions, keeping the research team abreast of relevant activities happening in the community, and problem-solving around implementation challenges experienced by the study (e.g., challenges with recruitment) and the CTCs (e.g., support to identify local resources that the CTC could refer to address specific client needs). The CABs met six times during the course of the study. Some CAB members also contributed to the interpretation of study findings and developing key messages [38].

## Study design

We used a qualitative descriptive study design [39] to identify facilitators and barriers to the implementation of the CAST intervention. We used the Standards for Reporting Qualitative Research (SRQR) to report study methods and results [40]. Data collection and analysis were guided by the Consolidated Framework for Implementation Research (CFIR) [41]. CFIR is a commonly used framework for examining health service implementation using evidence-based factors associated with successful implementation of interventions across a wide range of settings [42]. This framework consists of 39 constructs organized into five main domains known to affect implementation of interventions: (a) intervention characteristics, (e.g., adaptability, complexity, and design quality); (b) outer setting, which includes external factors (e.g., political, social, and economic); (c) inner setting (e.g., organizational factors such as structures, communication, and readiness for implementation); (d) characteristics of individuals involved in the intervention including their knowledge and beliefs about the intervention, stage of change, and personal attributes; and, (e) implementation process including planning, engagement of key players, and how the intervention is executed [41]. Use of the CFIR constructs can inform an understanding of why the implementation of an intervention was successful or unsuccessful [43].

## Participants

This study focuses on the implementation of CAST with older adults who received the intervention. Study participants were members of the three local CABs (a total of 35 individuals) and four CTCs. All CAB members and all CTCs were invited to participate in the study. There were no exclusion criteria. Client and caregiver perspectives on the implementation of CAST were also sought through post-intervention interviews. However, very few clients and caregivers agreed to take part in these interviews. As a result, these data were not included in the analysis.

## Data collection

**Document review.** We analysed study documents associated with the CAST intervention implementation to develop an understanding of and gain insights into implementation issues as they occurred at the time [44]. Data included meeting minutes for: (1) introductory and strategy (planning) meetings attended by the research team (principal investigators, co-investigators, study coordinators, research assistants, as applicable) and key study partners representing the local hospitals and community service organization; (2) CAB meetings, attended by

members of the research team and CAB members; (3) CTC meetings, attended by the principal investigators, study coordinators, and CTCs within each site; (4) recruiter meetings attended by research team members and hospital recruiters; (5) internal research team and co-investigator meetings; and (6) miscellaneous meetings held with study partners (e.g., hospital partners) regarding CAST implementation. Documents reviewed are summarized in S1 Appendix. A total of 233 documents were reviewed. Meeting minutes were recorded by the study coordinator or a research assistant.

**Interviews.** All CTCs were invited to participate in semi-structured individual interviews conducted at the end of the intervention period. CTC interviews were conducted by a research assistant (RA) either in-person or by phone between September 2018 and February 2019. CAB members were also invited to participate in two focus group interviews (one mid-way through the intervention implementation, and again at the end of the study recruitment period). CAB focus group interviews were conducted in person by research team members (RG, CW) between August 2018 and July 2019. All interviews were digitally recorded and transcribed. Transcriptions were checked against audio recordings for accuracy.

Interview questions were developed by the research team and guided by the CFIR framework [41] to describe the CAST intervention implementation across all three sites. Sample interview questions for each CFIR construct are available at http://cfirguide.org; as not all constructs were relevant, we selected questions associated with constructs most relevant to the CAST intervention. In addition, because the CFIR guides were developed for interventions implemented by an individual organization, and the CAST intervention involved multiple organizations, interview questions were adapted to reflect this difference. Others who have used CFIR to study the implementation of hospital to home transitions have also made adaptations to the framework to account for the involvement of multiple organizations [25].

During the first CTC interview, participants were asked questions regarding: (a) their perceived preparation for the role (training, provision of materials and support, confidence level in implementing the role and factors contributing to this); (b) facilitators and barriers experienced with implementing the intervention; (c) intervention changes or adaptions required to support implementation in their community; (d) support from and engagement of influential community members; (e) intervention complexity, including identification of components most successfully implemented and those most challenging to implement; and (f) potential outcomes for clients and caregivers. In the second CTC interview, participants were asked the same questions and were also asked to describe the support received from the research team. The guides for these CTC interviews are presented in S2 Appendix.

The focus group interviews with CAB members focused on their perceptions of how the CAST intervention was implemented in their communities. In the first CAB focus group interviews, participants were asked to describe: (a) their understanding of the CAST intervention; (b) why it was being implemented in their community; (c) how it compares to and would be integrated with existing local programs; (d) changes or adaptions required for the intervention to be effective in their community; (e) whether the intervention was complicated; (f) anticipated effectiveness in terms of its ability to meet community needs; (g) community response to the intervention; and (h) potential barriers to participation. In the second CAB focus group interviews, participants were asked about: (a) the status of intervention implementation; (b) their understanding of whether the intervention was implemented as planned; (c) sustainability of the intervention beyond the research study; (d) and their knowledge of intervention participants' experiences receiving the intervention. The guides for these CAB focus group interviews are presented in S3 Appendix. All interview participants (CTCs and CAB members) received the interview questions in advance to review.

## Data analysis

All anonymized interview transcripts and meeting minutes were uploaded to NVivo 12 [45] for data management and analysis. We used qualitative content analysis, consistent with a qualitative descriptive approach using deductive coding with the CFIR framework and inductive analysis for sub-codes within the CFIR domains [39]. This analytical approach allowed us to develop an understanding of the factors that served as facilitators and barriers to the implementation of the CAST intervention.

An RA, not involved with data collection, independently analyzed all interview transcripts line-by-line applying a priori codes of the CFIR domains and constructs. In addition, lower level nodes were developed inductively from the data and then integrated into the CFIR domains framework, as appropriate. Interview responses were coded across all CFIR domains. In addition to the RA, one principal investigator (CM) independently coded 20% of the interview transcripts to ensure inter-rater reliability in coding and then reviewed all subsequent coding with the RA to further categorize or collapse second level codes related to the CFIR constructs. Other authors (MMR, RG) also reviewed the coding within each of the CFIR domains. The research team met to review the coding to identify relationships between the codes and, where relevant, further combine or collapse codes, discuss emerging ideas, and identify discrepancies; adjustments to coding were made through consensus. The RA and lead author (CM) met to review all codes to identify patterns among first, second and third level codes [46], which were then merged into categories representing implementation factors. Themes and subthemes were then identified and reviewed by the research team, who agreed on data interpretation by consensus. The RA reviewed the data to select exemplar quotes; modifications and explanations within quotes are presented within square brackets to increase comprehensibility. Each quote has been identified as coming from CTC interviews or CAB focus groups in square brackets following the quote.

A similar approach for the analysis of study documents was used. Study documents were reviewed and analyzed by three authors (CM, MMR, RG). Data from the document analysis were triangulated with findings from the interview analysis.

We ensured study rigour with an audit trail of all procedures and decisions related to recruitment, data collection, and analysis. Interview transcripts and documents were reviewed independently to reduce potential for selection bias regarding positive responses and increase attention to negative responses. All authors were involved in providing feedback on the analysis and interpretation of findings.

## Ethical considerations

We obtained institutional ethics approval for this study from the Hamilton Integrated Research Ethics Board, McMaster University (REB#2586), the Office of Research Ethics at the University of Waterloo (#40867), the Laurentian University Research Ethics Board (REB#6009840), the Research Ethics Boards from Health Sciences North (REB # 17–007), and Joseph Brant Hospital (REB #000-039-17). All interview and focus group participants provided written informed consent prior to data collection.

## Results

### Participants characteristics

We conducted individual interviews with four CTCs across the three sites. These interviews were on average 35 minutes in length. All the CTCs were female, and were Registered Nurses with baccalaureate degrees; one CTC had a master's degree. The CTCs had experience working

in both acute care and community settings, with 2–20 years of clinical experience between them.

Two CAB focus group interviews were conducted at each site. At Time 1, 13 CAB members participated across the three sites; at Time 2, 12 members participated across the three sites. All but 1 focus group participant was female. In both Sites A and C, a CAB member was unable to attend one of the focus group interviews, but was interviewed individually at a subsequent time. The focus groups and interviews ranged from 35 to 87 minutes in length, with an average of 53 minutes. In Site A, 5 CAB members participated in the focus group/interview at Time 1 (2 community partners and 3 patient and public research partners) and 4 participated at Time 2 (2 community partners and 2 patient and public research partners). In Site B a total of 7 CAB members participated in the focus group at Time 1 (4 community partners and 3 patient and public research partners) and 3 participated at Time 2 (1 community partner and 2 patient and public research partners). In Site C, 1 CAB member (a community provider) participated in an interview at Time 1 and 5 (4 community partners and 1 patient and public research partner) participated in the focus group at Time 2 (S4 Appendix).

## Facilitating factors

The qualitative analysis of interview transcripts and program documents generated five themes that describe factors which facilitated the implementation of the CAST intervention. The themes were: (1) engaging the community to gain buy-in and adapt CAST to the local community context; (2) planning, training and research meetings; (3) engagement, relationship building and collaboration with local partners; (4) availability and provision of support, information and resources to address CTC needs; and (5) tailoring the intervention to individual client needs and preferences. The CFIR domains and constructs that contributed to each theme are described.

**Engaging the community to gain buy-in and adapt CAST to the local community context.**   The first theme identified was engagement with each of the participating communities and how this helped to facilitate the implementation of the CAST intervention. Three CFIR domains (constructs) contributed to this theme: Process (Planning), Intervention Characteristics (Adaptability), and Inner Setting (Implementation Climate). This theme, which included two subthemes, highlighted the importance of gaining buy-in within each community and adapting the implementation of the CAST intervention to the local context. This required an understanding of current services and supports for hospital to home transitions in each community, where gaps existed, and how CAST could be adapted to fill those care gaps. The importance of engaging the individual communities is highlighted in the following quote:

*"Obviously you're doing the right thing by asking the community, asking: 'What do you think is needed? What is the gap her?' Because some places [. . .] might have something for a certain issue . . .why re-invent the wheel? [. . .] but I think you're doing the right thing, because you're asking. 'What is it you need? Where is the gap?'"* [CAB Site C]

## Introducing CAST and obtaining local buy-in

The document analysis revealed the sub-theme–introducing CAST and obtaining local buy-in. Within each community, attempts were made to understand the local context in terms of existing services, the priority of the intervention, and alignment of CAST within participating hospitals' strategic plans, which supported community buy-in. Understanding the local context began with holding a Community Forum in each site prior to study initiation.

The forums enabled researchers to learn more about community resources, including existing programs and services that supported the transition of older adults from hospital to home. The Community Forums also provided the community with an opportunity to learn about the CAST intervention implementation and evaluation plans. Input from the Community Forums helped to identify members for the local CABs that were subsequently established to support local implementation of CAST. Together, the Community Forums and CABs provided the research team with information on the unique characteristics and needs of each community. Meeting notes also highlighted the variability of services available in the different study communities, with one community having fewer community services available to clients following hospital discharge than were available in the other communities. Meeting notes reflected sensitivity in understanding how the CAST intervention, as a new service in the region, implemented by an external research group, would be perceived.

## Adapting the implementation of the intervention to the local context ensured that CAST complemented existing services

Understanding the diverse characteristics and needs of the local context and adapting the implementation of the intervention, helped to ensure that the intervention complemented and did not duplicate or interfere with existing services. A CAB member described the importance of adapting the implementation to the local context to help with service efficiency and reducing duplication:

*"Groups will be able to share all of the processes, and because of that, it will improve the efficiency of the work that has to be done, because the more resources, the more that can be shared. And, the first thing they'll have to do is look at where there's duplication of services, and take out that duplication and focus on the priority things that need attention, through all of the resources that each of the groups will have."* [CAB Site C]

Given that other services in each community also help to support older adults transitioning from hospital, there was a potential that services could be seen as competing. Adapting the implementation for each community helped to prevent this. One CAB member highlighted the significant needs of this population and that there were no feelings of competition in their community as a result of working together.

*"We can divide and conquer and support people because there's a lot of vulnerable people in our city, and it takes a village, right? So, I don't feel any territorial things at all."* [CAB Site C]

Meeting notes also revealed that there was discussion about adapting the CAST intervention to meet local needs within each of the three communities. Fewer adaptations of the intervention were made in one community which did not have a vast array of available community services. One concern raised during the Community Forums was about the transition back to usual care following receipt of the CAST intervention. To address this, CAST was adapted to enhance the discharge process at the end of the 6-month intervention period. A clear plan was developed and documented, outlining which organization(s) participants would be referred to (where applicable) after completing the intervention.

**Planning, training, and research meetings.** The second theme identified as facilitating the implementation of the CAST intervention was planning and training meetings as well as ongoing research meetings with the CTCs. Four CFIR domains contributed to this theme: Intervention Characteristics (Relative advantage, Complexity, Design, quality and packaging);

Inner setting (Readiness for implementation–access to knowledge and information); and Process (Planning). There were two subthemes identified within this theme; the subthemes reflected the extensive training provided to the CTCs to prepare them for their roles and the ongoing support provided by the study principal investigators and study research coordinator.

## Training of the CTCs

Notes from research team meetings described the development of a comprehensive, standardized training manual. The manual included: (a) information on the study population; (b) resources to support the CTCs in implementing each component of the intervention (e.g., clinical assessment tools, medication reconciliation tools, problem-solving therapy); (c) research articles and presentation notes to support both the rationale for, and implementation of the intervention components; (d) documents to use when communicating with other providers (e.g., family physicians); and (e) tools to document CTC activities. To further support the CTCs, the CTC who was the first to begin working in the role supplemented the training manual with additional resources that she found or created to assist her in working with clients and caregivers.

Training was provided over a 2-day period, led by the study principal investigators and supported by the study's research coordinator and a CTC (except when the first CTC was trained). As part of the training, CTCs engaged in discussions about the intervention rationale and components, watched training videos, and had opportunities for role play. Training notes revealed that CTCs valued time spent discussing their nursing role, specifically related to their role in case management, care coordination and system navigation support, and their experience developing interorganizational relationships.

## Ongoing and timely access to the research team by the CTCs

Meeting notes indicated that the regular research team meetings with the CTCs while they implemented their role served as an opportunity to: (a) address questions of a clinical nature (e.g., clinical supervision of CTCs, management of clients that became emotionally distressed during visits, management of suspected elder abuse); (b) reinforce and address questions related to implementation of the study (e.g., participant recruitment, documentation, ethics/confidentiality); and (c) share success stories related to positive clinical outcomes and effective collaborative efforts with community partners. These meetings also served to discuss strategies to address clinical issues within the context of local resources and supports for clients available within each site. (e.g., how to meet client needs for specific supports, such as physiotherapy or exercise programs, given limitations in availability or eligibility criteria for these services). The CTCs valued prompt access to the principal investigators via phone, email and in-person to address implementation related questions. The following two quotes highlight the support provided to the CTCs:

> "So we've had regular CTC meetings with the study [principal investigators] over the course of the intervention to touch base on what's working well and what may be not working so well, as well as study [principal investigators] and research coordinators have been able to be reached by phone or email or in-person at the office as needed for any further direction."
> [CTC Site C]

> "Well I think, having the study [principal investigators] present [for training] was positive for sure because I had the opportunity to digest the information, come back the second day and ask some follow up questions. Since then things have been evolving a bit and there's been some, ongoing questions and not retraining but follow-up for sure. . . . And having the

*[principal investigator] in the office available in person or by phone for specific concerns with intervention visits, I have to bounce off someone and clinical guidance for any unexpected situations so that's been good to have that."* [CTC Site B]

**Engagement, relationship building, and collaboration with local partners.** The third theme that was identified as a facilitator to implementing the CAST intervention related to engaging, building relationships, and collaborating with health and community service partners in each of the participating sites. Four CFIR domains contributed to this theme: <u>Intervention Characteristics</u> (Adaptability); <u>Process</u> (Engaging–Opinion Leaders, Formally Appointed Internal Implementation, Champions, External Change Agents; Executing); <u>Outer Setting</u> (Cosmopolitanism); and <u>Inner Setting</u> (Networks and Communication). This theme included two subthemes, the emergence of local champions who helped to support and promote the CAST intervention, and collaborations with local providers to support and enhance client care.

## Local champions supported and promoted the implementation of the intervention

In the process of developing and implementing CAST, some individuals representing hospitals and community services (e.g., senior nursing leaders) with whom the CTCs initially met, became effective champions of the intervention. These champions supported the intervention within their organizations, which served to legitimize the work of the hospital-based recruiters as well as the CTCs. Meeting notes reflected that, within hospitals, intervention champions promoted the study, helped identify eligible clients, and facilitated recruitment by increasing access to various hospital units. One CAB member described the value of having the support of a vice president at the hospital to the implementation of the intervention:

*"And I think* [Hospital Vice President] *too, having someone who's high-up in the hospital, she was also a really strong advocate for this. . . to me that was something that was very beneficial from my perspective."* [CAB Site C]

Similarly, in the community, organizational leaders who were champions of the CAST intervention provided staff members in their organization with information about the CAST intervention and advocated for the importance of assessing referrals from the CTC. An example of a community-based champion was highlighted by a CAB member:

*"Having* [the partner] *go in and talk to people and say "This is not just another box for you to check every day, another job for you to do", this is a really important thing for the community."* [CAB Site C]

Document analysis revealed that some champions also helped identify and connect the CTCs with leadership from other services to further support implementation.

## Established relationships with health and social services facilitated information sharing and collaboration for client care

Document analysis revealed that early in the development of the intervention, CTCs scheduled meetings with program leaders (such as managers from home care) to explain the intervention and explore synergies in service delivery and opportunities for collaboration. These meetings

resulted in establishing working relationships that optimized the CTCs' clinical care of clients through enhanced information sharing across services, expedited referrals to community services, and enabled collaborative problem-solving to match services to specific client needs. One CTC provided an example of how she worked collaboratively with another provider to meet the goals of a client:

> *"I've made a few calls to care coordinators, saying you know there was this concern in the home but this is their goal, how can we—can you think of anything I can do in the home to help them meet this goal safely and please let me know. And I receive a call back saying let's try to modify it this way and I said okay great, I'll provide that same encouragement so everyone's on the same page."* [CTC Sites C]

Having strong relationships with other providers also helped the CTCs understand the work realities of other providers. This CTC acknowledged the limited time available to some home care providers and described the appreciation that home care providers expressed in being able to work together to meet the client's needs:

> *"So [the care coordinators from Home and Community Care are] spread very thin. So going in to reassess, they don't have the opportunity to do that as often in some cases and . . . perhaps [see] other needs that are . . . changing over time . . . the care co-ordinators that I've spoken to, they appreciate [me] at least connecting with them and sending my notes, providing these updates . . . because of their caseloads and their time limitations. . ."* [CTC Site B]

Another CTC explained working with a physician to involve another health care professional to support the care needs of a client:

> *"We worked with a physician to get a social worker involved with my case. That we were trying to get* [client] *into a complex continuing care facility. . .. And the nurse that was working with me and the social worker that was working with me were all on board to get her there."* [CTC Site A]

**Availability and provision of support, information, and resources to address CTC needs.**   The fourth theme that was identified as facilitating the implementation of the CAST intervention addressed the support provided to the CTCs to undertake their role. Three CFIR domains contributed to this theme: Intervention Characteristics (Complexity); Inner Setting (Networks and Communication, Readiness for Implementation–Available Resources, Access to Knowledge and Information); Process (Planning). Two subthemes included the support, information and resources provided to the CTCs and the establishment of a community of practice (CoP) to provide opportunities for the CTCs to discuss cases and problem-solve together.

## Provision of resources based on CTC needs for information

The review of meeting documents revealed that CTC need for information and supports were regularly assessed in meetings and resources were developed or existing resources were tailored to address these needs. These resources included consent forms to allow for communication within the circle of care, information on problem-solving therapy, behavioural change support, system navigation, behavioural and discharge planning to facilitate linkages and transitions across services at the end of CAST program, and creating hard copies of client and

caregiver resources for education. The CTCs also described the importance of having access to information on available community services in terms of what services were provided, eligibility criteria, and referral forms and processes for access. Two CTCs explained the value of this information:

*"So the community resources were available to me. All the different community resources. All the forms were available to me. All the supporting data on how to do it was available to me."* [CTC Site A]

*"Getting information on day programs too was, was important just to understand how those work.."* [CTC Site B]

### Development of a CTC community of practice

To provide additional support to the CTCs, the research team established a CTC CoP, that included the CTCs from each of the sites and the study research coordinator. The principal investigators attended the initial meetings of the CoP and later attended when invited by the CTCs. Document analysis revealed that through the CoP, CTCs discussed questions and issues related to intervention implementation, and shared effective strategies for managing these issues, and brainstormed solutions for common problems. When new CTCs were recruited, established CTCs offered to provide training and mentorship support, for example, attending home visits together (job shadowing) and offering key insights gained through implementation of their role. One CTC described the value of this peer-to-peer support:

*"My colleague* [CTC in another community] *is available to help discuss cases, work through how to implement components of the intervention. We talk about clinical questions with each other. . . So the most supportive thing that works for me is to be able to discuss individual patients at an individual level with* [CTC]." [CTC Site C]

**Tailoring the intervention to individual client needs and preferences.** The final theme that facilitated the implementation of the intervention involved the person-centred focus of CAST which enable the CTCs to tailor the implementation of the CAST intervention to address individual client needs and preferences. This theme was based on one CFIR domain: <u>Intervention Characteristics</u> (Adaptability, Relative advantage). A CAB member provided one example of the person-centred implementation of CAST; specifically, respecting the preferences of the clients in terms of how and how often the intervention was delivered:

*". . .talking to the client and saying 'Okay do you want me to call you once a month?. . . How [*do*] you see this going forward?. . . Would you prefer I come to the house?. . . working together and having it be client-centered or client-driven in terms of the length [*of meetings*]."* [CAB Site C]

Tailoring the intervention also involved supporting clients to determine what aspects of their care they want–and are ready–to focus on. A CTC explained:

*". . .for me as a nurse seeing needs but [*the clients*] don't want to engage in that . . . but again trying to be respectful of their readiness to change or . . . maybe their cognitive status in that they just weren't ready or willing to engage at that time . . . not all the participants wanted or*

*were able to accept all the different components [of the CAST intervention], but [offering the components at] each visit and then seeing where they are at that time was my strategy."* [CTC Site B]

The structure of the intervention, which allowed the CTC an opportunity to get to know each client over an extended time period, contributed to the CTC's ability to provide more person-centred care. A CTC described the difference between CAST and other programs:

*". . .in terms of the support, it appears to me that it's very client-centred and one-on-one, and like that. The nurse gets to know the people. Versus, you know, some other programs where there might be huge caseloads or what. So, I think the role is very targeted and defined."* [CAB Site B]

The analysis of documents from meetings with the CTCs highlighted the importance of delivering the intervention based on client goals and preferences for care. The CTCs described the importance of working toward goals identified by clients as this could ultimately impact depression. For example, focusing on the attainment of a client's mobility goals and advocating for physiotherapy could result in improved quality of life and mental well-being.

**Barriers to implementation.** The qualitative analysis of the interviews with CTCs, CAB focus groups and program documents generated four themes that described barriers associated with CAST implementation. These included: (1) difficulty recruiting and retaining intervention staff; (2) difficulty engaging older adults in the intervention; (3) balancing tailoring of the intervention with ensuring core components are delivered; and (4) difficulties engaging clients' circles of care made care coordination challenging. The CFIR domains and constructs that contributed to each theme are described

**Difficulty recruiting and retaining intervention staff.** The first theme identified described the difficulties recruiting and retaining staff members to implement CAST. This theme was based on one CFIR domain: Process (Executing). Document analysis revealed significant challenges recruiting nurses to assume the CTC role. Much meeting time was devoted to the discussion of recruitment issues and identifying potential recruitment strategies. Study partners in each community were engaged early in the planning process to identify potential candidates. This included contacting professional nursing associations (e.g., Registered Nurses Association of Ontario). Outreach was also conducted with existing community services. Restructuring of the administration of home care services in the province, which took place at the time of the study, was identified as a potential opportunity to identify available nurses for the CTC role. Study and community partners were canvassed to consider the potential for a one-year secondment of their staff. However, local stakeholders noted that recruitment of nurses was a common problem, describing it as a human health resources crisis. Moreover, it was documented that within each setting there were unique considerations as to which study partner organization would be involved in the staff hiring process. In fact, CTCs were hired by a university in two of the sites and by a hospital in the third site. Implementation considerations included challenges associated with hiring into unionized workforce environments (e.g., needing to be open to hiring other types of clinicians, different established salary ranges, and the inability to post the position as a non-union position because it involved direct clinical work). Given the active involvement of study partners in the recruitment process, the research team wanted to ensure that they were also involved in the interview and final hiring decision-making process. Keys issues that impacted recruitment of potential individuals was the time-limited length of the contract, and that the position was full time when, in at least one case, there was an interest in part-time employment. As recruitment of a full-time CTC in one

community proved to be extremely difficult, the position was changed to part time/temporary/casual to increase the likelihood that the position would be filled.

**Difficulty engaging older adults in the intervention.** The second theme that was identified related to challenges in engaging clients in the intervention. This theme was based on two CFIR domains: Intervention Characteristics (Complexity) and Process (Executing). While most clients were glad to have the support of the CTC and open to working with the CTC, other clients were more difficult to engage. Some clients did not want support, indicating that they could manage on their own. As one CTC explained:

> "...the person that's like nope, no needs... We're managing, we have this..." [CTC Site B]

Others were not ready to engage in certain aspects of the intervention, such as discussing mental health issues. The CTCs explained that for some clients, stigma associated with mental health conditions was an issue:

> "There is the stigma where people don't always want the information on depression, or aren't willing to learn more at some stages especially if they are saying, no my mood's okay, but their geriatric depression scale says it's not great but they are still not in the mindset of getting that information." [CTC Site B]

The CTCs explained that other clients already had numerous supports in place, provided either by family or through a retirement home where they lived, and thus did not appear to need assistance from the CTC. For example:

> "...I'm finding in my area [city], I'm getting a lot of people that live in assisted living or retirement homes. So when they hear that what they are actually going to receive as part of the intervention is a nurse to help coordinate care, to help review and manage medication, or help them discuss their mood or set goals, many of them are very busy with their retirement home activities, and they are also very connected . . . and I have a lot of people on my list that have declined." [CTC Site C]

**Balancing tailoring of the intervention with ensuring core components are delivered.** The third theme that was identified related to the CTCs trying to balance providing all components of the intervention while also tailoring the intervention for individual clients. This theme was based on one CFIR domain: Process (Executing). As the intervention was intended to be person-focused, the CTCs experienced conflict when client needs were not consistent with the components of the intervention that were expected to be implemented as part of the research study. For example, there were expectations for assessment tool administration or use of specific strategies such as problem-solving therapy that the CTC may not have perceived as clinically relevant for the client. In these situations, the CTCs struggled with maintaining the intervention as person-centred, and, in some instances, intervention components that should have been implemented according to the intervention plan were not implemented. Research team meeting notes also identified this challenge noting that the CTCs tried to find this balance. One CTC explained the struggle to find a balance:

> "It's challenging to get in all the intervention components and to, looking at what's client-driven during the visit, they might have different priorities than me doing all the screening

*tools, for example. . . And so far, I've really had to follow the direction of the participants, which is not always in line with including all the intervention components. . ."* [CTC Site B]

Problem-solving therapy was one component of the intervention that CTCs identified as being particularly challenging to implement. One CTC described the challenges some clients had with problem-solving therapy due to its abstract nature:

*"Yah, problem solving therapy has been challenging with this population; it's very hard for some people to identify the problems they want to work on in a formal sense, some are not cognitively able to engage cause it's more abstract and they might be more concrete with their thinking, or not you know ready emotionally to engage. . ."* [CTC Site B]

**Difficulties engaging clients' circles of care made care coordination challenging.** The final theme that served as a barrier related to the challenges engaging a client's circle of care and the impact of this on coordinating their care. Two CFIR domains contributed to this theme: Inner Setting (Networks and communication) and Process (Executing).

Care coordination was a key aspect of the CTC role. To do this effectively, an initial step in the intervention was to identify the client's circle of care and reach out to these providers to understand what services the client was receiving and to establish relationships with them. Within this theme, two subthemes described challenges engaging providers in the client's circle of care and engaging family physicians.

## CTCs engaging with providers in the client's circle of care

Care coordination was challenged by the fact that the CTCs lacked information about who was in the clients' circle of care. Many clients, while aware that services were provided in their home, did not always know who these care providers were or what agency they were from. A CTC described this challenge:

*"The other issue that comes up often is determining the circle of care; and then what, who is the most important to communicate with in the community, so that you can appreciate a lot of people don't fully appreciate who's in their circle of care or who is coming or where they are from; so they sometimes mix up organizations so that's been a challenge too."* [CTC Site B]

CTCs reported spending a lot of time identifying service providers involved in clients' care and experienced difficulty connecting with them. The CTCs did not have access to the clients' electronic medical record so they were unable to identify community providers involved in their care through this administrative mechanism. Further, there was no shared information system for all of the providers and agencies involved, which further challenged care coordination. A CTC explained the multiple steps involved in trying to identify the circle of care, and the added complication of not having access to a shared information system to assist with this work:

*"I think it's quite complicated . . . trying to connect with the circle of care at times. Figuring out who the LHIN care coordinator is, connecting with the team, faxing them, calling them back to make sure that they did get the fax, and the other notes. So unfortunately there was no streamline[d] set of communication process, not having access to clinical connect* [a web-based portal for sharing health information within the health care system] [CTC Site C]

## Engaging family physicians

CTCs noted that they had minimal interactions with clients' family physicians despite attempts to connect with them via phone and fax. Communications were described as mostly one-way, with CTCs sharing their concerns and observations about clients, but receiving few responses from family physicians. A CTC explained her experience:

*"I didn't do a lot of connections [with family physicians]. It seemed mostly by fax. And I have done follow-up calls . . . mostly alerting . . . maybe some medication concerns here or . . . their goal for maybe a referral to a program, alerting them of any depressive symptoms. We did get some notes back and communications back but it was mostly one way."* [CTC Site C]

It was also documented in the meeting notes that CTCs reported difficulty getting information from family physicians on the client's medical history to support their work (e.g., information on diagnoses and current medications). It was suggested that this may be the case, in part, because family physicians did not know whether they could trust the CTC as they did not have established working relationships or because clients have so many home care services that physicians were not sure who the CTCs were or with which service they were associated.

CTCs described the importance of being able to engage with family physicians and the need to have them actively involved. For example:

*"Well we have to get more physicians, the family physicians, to be on board. And be open to receiving that feedback that we're seeing in the community. The primary care providers of patients. . .They were not receptive to feedback."* [CTC Site A]

## Discussion

This study aimed to identify the facilitators and barriers encountered during the implementation of a new nurse-led intervention for older adults with multimorbidity and depressive symptoms transitioning from hospital to home. The factors identified were informed by the following CFIR domains: <u>characteristics of the intervention</u>, the <u>inner setting</u> (context within which the intervention was implemented), <u>outer setting</u> (the broader economic, social and political context that could influence the intervention), and the <u>processes</u> required for implementation. Thus, this research enhances our understanding of what is required to successfully implement transitional care interventions, particularly for populations with complex care needs such as older adults with multimorbidity and depressive symptoms.

Implementation facilitators included working collaboratively with the participating communities to establish relationships, gain buy-in for the intervention, and adapting the intervention to fit the individual community context. Early engagement of stakeholders and those likely to be impacted by an innovation have been identified as important to obtaining support and increasing the likelihood of intervention success [41, 47]. However, the time required to develop these relationships between the research team and local stakeholders is significant and ample time should be built into project timelines to enable these relationships to be nurtured and trust established.

CAST was designed to be a person-centred, multi-component intervention where the needs and preferences of individual clients informed the activities and priorities of the CTCs. The CTCs had the flexibility to spend time getting to know the client, engaging with them in identifying their goals of care, and then working with their clients on developing care plans. In a systematic review of patient-centred care and multimorbidity, person-centred care approaches

were found to be strongly associated with positive health outcomes [48]. Eight elements that were identified as being part of a person-centred care approaches, informed the development of the CAST intervention: (a) frequent contacts with clients; (b) conducting face-to-face clinical interactions; (c) developing care plans and interventions based on client needs and preferences; (d) incorporating the needs of family members in the supports provided; (e) sharing the plan of care with the client and other providers; (f) making referrals to needed services and supports; and (g) providing the client with feedback [48]. Person-centred approaches to care are therefore important in terms of improving health outcomes and in helping to facilitate the implementation of an intervention.

While the person-centred nature of CAST was identified as a facilitator in the study, certain aspects of the intervention were considered challenging to implement. Problem-solving therapy, a core component of the intervention, was identified as particularly difficult by all the CTCs. The CTCs indicated that they routinely engaged in problem-solving with clients but reported that it was difficult to undertake a more systematic approach to addressing issues faced by clients because of the time required. In addition, some clients were not receptive to exploring problems they were having with their mood and developing plans to make improvements in this area. While problem-solving therapy has been shown to be an effective treatment for older adults with depression [49, 50], attention needs to be paid to individual (e.g., motivation) and logistical (e.g., time) factors if practitioners plan to incorporate problem-solving therapy as part of a multi-component intervention for older adults with multiple health issues.

Most studies on transitional care interventions have not included populations most susceptible to health challenges associated with fragmented transitional care, such as older adults with multimorbidity and depressive symptoms [51–55]. Thus, the CAST study served to address this gap in the literature. The challenges associated with care coordination for individuals with multimorbidity have been highlighted by others [56, 57]. Because of the multiple chronic conditions that they live with, the circle of care for individuals in this population are often broad and complex, and individuals might not always know which providers are in their homes and the organizations that they are affiliated with. CTCs spoke to the challenges of identifying and engaging with clients' circles of care, with family physicians being a particularly challenging group to engage. The involvement of primary care physicians in community-based health interventions are essential since they serve as care coordinators and in some instances gatekeepers to other services [58, 59]. Other studies have also identified challenges with engaging primary care providers in health care and system innovation [59, 60]. Barriers to engagement include: reluctance to sharing patient-level information, funding models, legal concerns, care complexity, and local physician shortages [61–63]. Early engagement of primary care providers to understand and address context-specific barriers should be undertaken to increase the likelihood of meaningful physician engagement.

Challenges were also faced in terms of recruiting RNs to serve as the CTC, particularly in one of the participating communities where a health human resource shortage existed prior to study initiation. The research team worked with multiple stakeholders and explored creative strategies (e.g., secondments, utilization of retired nurses) to recruit CTCs. Challenges of recruiting and retaining nurses have been cited by others [64] and are particularly acute in rural areas [65]. Ideally, CTCs in a transitional care intervention would have a good understanding of both hospital and community care systems, including experience conducting home visits, and have experience working with the target population, in this case older adults with multimorbidity and depressive symptoms. Nurses meeting these criteria likely have greater years of experience and would be working in senior positions and, therefore, may be less interested in taking on a time-limited position as part of a research study. Establishing

secondments for nurses who are already working in the system may help to enhance recruitment efforts.

Engagement of diverse stakeholders in community-based health interventions is challenging but essential, especially for interventions targeting complex older adults with multimorbidity transitioning from hospital to home. To foster and encourage such collaborations, policymakers should explore approaches to support these collaborations, including interventions that enhance quality of care of patients and improve working conditions and experiences for health care providers. Other incentives such as remuneration may also be effective [47, 66].

## Strengths and limitations

A strength of the study was use of the CFIR framework to guide the development of data collection tools as well as the analysis. Using CFIR to guide the study enabled us to build on the existing knowledge about implementation science to explore determinants or factors at multiple levels of influence on implementation of a transitional care intervention for older adults with multimorbidity and depressive symptoms. However, one challenge with using CFIR was that it was developed to examine the implementation of interventions by individual organizations. As CAST was a community-based intervention, the data collection tools needed to be revised to reflect the community-level focus of the intervention. Another strength of the study was that it was conducted in three diverse geographic regions, enabling the opportunity to explore factors that helped or hindered implementation of the intervention in multiple settings. In terms of limitations, while data collected from the CABs was a source of data in the study, we did not obtain information from a broader range of community members regarding the implementation of CAST. As well, since CAB members were 'partners' in the project, we did not collect information on their demographic characteristics, limiting our ability to describe these groups. Finally, while attempts were made to gather information from clients and caregivers regarding the implementation of CAST, because of the complex health conditions of the clients (and the impact of this on caregivers) very few agreed to take part in this component of the study. As a result, this study is missing this important perspective.

## Conclusion

The study identified facilitators and barriers that impacted the implementation of a nurse-led community-based transitional care intervention for older adults with multimorbidity and depressive symptoms. The findings suggest that considerable time and effort is required to work collaboratively with participating communities to ensure there is buy-in and commitment, to engage client's circle of care to ensure coordinated and comprehensive care is delivered in ways that meet clients' needs, and to appropriately support those delivering the intervention. Investing in these relationships and activities can help contribute to successful intervention implementation.

## Supporting information

**S1 Appendix. Meeting minutes reviewed for document analysis.**
(DOCX)

**S2 Appendix. Interview guides for CTCs.**
(DOCX)

**S3 Appendix. Focus group guides for CABs.**
(DOCX)

**S4 Appendix. CAB focus group and interview participants.**
(DOCX)

## Acknowledgments

We are grateful to the individuals who participated in this study, the communities that agreed to trial the CAST intervention, and the many partners who provided support and assistance throughout the project. We are also thankful to the Community Transition Coordinators, the site recruiters and research assistants, and the members of the Aging, Community and Health Research Unit in the School of Nursing at McMaster University, Hamilton, Ontario, Canada. We also extend our thanks to the members of the Community Advisory Boards for their contributions to the study: Nicole Blais, Rebecca Bowes, Monica Bretzlaf, Sarah Denton, Denise Freylejar, Rob DiMeglio, Fran McMillan, Jenn Osesky, Melanie Paul, Rebecca Poulin, Melissa Roney, Louise Trudel, Jennifer Turgeon, Mary Buzzell, Genevieve Giglia, Sue Gilbert, Gloria Jackson, Penelope Petrie, Shelley Wright, Hanadi Almasri, Sherrie Cheers, Trish Corbett, Joan Gallagher-Bell, Michael Hourigan, Joan Lewis, Karen Stein, and Nora VanDelan.

## Author Contributions

**Conceptualization:** Carrie McAiney, Maureen Markle-Reid, Rebecca Ganann, Ruta Valaitis, Kathryn Fisher, Jenny Ploeg.

**Formal analysis:** Carrie McAiney, Maureen Markle-Reid, Rebecca Ganann.

**Funding acquisition:** Carrie McAiney, Maureen Markle-Reid, Rebecca Ganann, Ruta Valaitis.

**Investigation:** Rebecca Ganann, Carly Whitmore.

**Methodology:** Carrie McAiney, Maureen Markle-Reid, Rebecca Ganann.

**Project administration:** Carrie McAiney, Maureen Markle-Reid, Diana J. Urajnik.

**Resources:** Carrie McAiney, Maureen Markle-Reid, Rebecca Ganann.

**Supervision:** Carrie McAiney, Maureen Markle-Reid, Diana J. Urajnik, Janet E. McElhaney.

**Validation:** Carrie McAiney, Maureen Markle-Reid, Rebecca Ganann.

**Writing – original draft:** Carrie McAiney, Carly Whitmore.

**Writing – review & editing:** Carrie McAiney, Maureen Markle-Reid, Rebecca Ganann, Carly Whitmore, Ruta Valaitis, Diana J. Urajnik, Kathryn Fisher, Jenny Ploeg, Penelope Petrie, Fran McMillan, Janet E. McElhaney.

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
