## [Decision Letter · Decision Letter 0]

21 Dec 2021

PONE-D-21-07160Implementation of the Community Assets Supporting Transitions (CAST) transitional care intervention for older adults with multimorbidity and depressive symptoms: A qualitative descriptive studyPLOS ONE

Dear Dr. McAiney,

Thank you for submitting your manuscript to PLOS ONE. After careful consideration, we feel that it has merit but does not fully meet PLOS ONE’s publication criteria as it currently stands. Therefore, we invite you to submit a revised version of the manuscript that addresses the points raised during the review process.

I have accepted the invitation to act as New Academic Editor as the original editor was no longer available. Considering the time the manuscript has spent in review, and the fact that the available review is very thorough, I have decided to invite you to revise the manuscript based on this review. I agree with the reviewer comments and recommend you address these comments in the revision. I added a few comments which need to be addressed with the revision. If you don't agree with a comment, please provide a justification in the response to reviewer comments.

We look forward to receiving your revised manuscript.

Kind regards,

Anja K Leist, Professor Dr.

Academic Editor

PLOS ONE

Additional Editor Comments (if provided):

While I understand that the focus of this study is the implementation of the intervention among professionals, stakeholders and the community at large, the missing perspective of the care recipients needs to be presented and justified already in the introduction. I would suggest to restructure the introduction to more focus on implementation issues instead of focusing on the care recipients. I agree with the review to also present some descriptive statistics (age, nature and/or number of chronic conditions) on the target or actual population. Were there any inclusion or exclusion criteria to be enrolled in the intervention?

The methods section mentions that the intervention was tested in a randomized controlled trial, but I didn't see any details on how the care recipients were assigned to intervention or control arm, and any comparison between intervention and control group. I recommend to add an explanation that this study focuses on the intervention group and the experiences with the implementation of the intervention.

I suggest some streamlining throughout the manuscript and particularly the discussion, as several details of the intervention and themes identified are repeated. The description of the intervention in the first section of the introduction and the study overview should be presented in the methods section.

Journal Requirements:

Reviewers' comments:

Reviewer's Responses to Questions

**Comments to the Author**

1. Is the manuscript technically sound, and do the data support the conclusions?

Reviewer #1: Partly

2. Has the statistical analysis been performed appropriately and rigorously? 

Reviewer #1: N/A

3. Have the authors made all data underlying the findings in their manuscript fully available?

Reviewer #1: No

4. Is the manuscript presented in an intelligible fashion and written in standard English?

Reviewer #1: Yes

5. Review Comments to the Author

Reviewer #1: INTRODUCTION:

Ensure that there is detail provided about the context for where the study occurred, defining key terms and the importance of this topic for older adults. Clarify and describe the type of intervention, such as is it an enhanced home care program and how it is similar to or different from home care in the province.

line 113: “initiated a coordinated care with these individuals” should read “initiated a coordinated care plan”

line 123: Be specific to describe what type of education (i.e., health, health services, self-management)

METHODS:

line 210 Participants: include the inclusion and exclusion criteria and move the description of the participants to the first part of the results.

line 213: This paragraph is confusing. Please provide a table with details of CAB and CTC.

line 325: Include a brief description about the target older adult population. It is important to identify to the reader the type and severity of chronic disease and severity of depressive symptoms the CTC were providing services for.

RESULTS:

First include a clear description of the characteristics of the study participants.

The use of the CFIR framework is being used as a deductive framework. Thus, the five main domains should be clearly utilized to structure the results. Provide more clarity between the domains and how the themes/subthemes related to each domain. Tables 1 and 2 are not necessary as they do not add new information. This should help to reframe the results deductively within the CFIR domains.

Ensure that there is a more thorough narrative description of the results in relation to each domain/theme/subtheme.

Include a clear connection between the narrative description of the findings and illustrative quotes.

Throughout the results, the term client is used. Clarify if you are referring to patients or caregivers, or both patients and caregivers

DISCUSSION:

Ensure that the key results are highlighted and clearly connect to the results of this study. For example, discuss the finding related to challenges in engaging family physicians.

Ensure that the key results are not just repeated but are discussed in relation to existing evidence. Ensure that the findings are interpreted in light of what the study adds to our current knowledge and/or how it confirms prior knowledge.

Include the implications of this research for policy and/or practice.

6. PLOS authors have the option to publish the peer review history of their article (what does this mean?). If published, this will include your full peer review and any attached files.

Reviewer #1: No

---

## [Author Response · Author response to Decision Letter 0]

16 Jun 2022

1. INTRODUCTION:

Ensure that there is detail provided about the context for where the study occurred, defining key terms and the importance of this topic for older adults. 

** In the feedback from the Editor, it was suggested that the Introduction be revised to focus more on implementation as that is the focus of the paper. We have kept much of the background section to provide the important context regarding transitional care but have added a paragraph related to implementation and the importance of understanding factors that influence implementation. 

2. Clarify and describe the type of intervention, such as is it an enhanced home care program and how it is similar to or different from home care in the province.

**It was also suggested that the description of the CAST intervention and overview of the pragmatic trial be moved to the Methods section. We have done this and within the description of the CAST intervention have provided more information about relationship between CAST and home care (specifically, that CAST is intended to complement home care and other services, and that clients receiving home care can also receive services from the CAST CTC).

3. Line 113: “initiated a coordinated care with these individuals” should read “initiated a coordinated care plan”

** This change has been made.

4. Line 123: Be specific to describe what type of education (i.e., health, health services, self-management)

** The type of education provided has been specified.

5. Line 210 Participants: include the inclusion and exclusion criteria and move the description of the participants to the first part of the results.

** We had initially decided to include some descriptive information about the CAB members and CTCs in the Methods (starting at Line 210). However, we realize that this may be confusing, so we have removed this information. We have also revised the manuscript to indicate that all members of the CABs and all CTCs were invited to participate in this study, and that there were no exclusion criteria.

6. Line 213: This paragraph is confusing. Please provide a table with details of CAB and CTC.

** As noted above, we agree that this information is confusing. We have removed the section of the manuscript that provided an overall description of the CABs and CTCs. A description of the CAB members and CTCs who participated in the study is provided at the beginning of the Results section, under Participant Characteristics.

7. Line 325: Include a brief description about the target older adult population. It is important to identify to the reader the type and severity of chronic disease and severity of depressive symptoms the CTC were providing services for.

** The eligibility criteria for older adults in the clinical trial are outlined in the Overview of the Pragmatic Trial in the Methods section.

8. RESULTS: First include a clear description of the characteristics of the study participants.

** A description of the characteristics of the study participants (CAB members and CTCs) is provided in the Results under Participant Characteristics. Note: Because the CAB members were working with us as partners to help implement the study, limited information about their characteristics was collected. 

9. The use of the CFIR framework is being used as a deductive framework. Thus, the five main domains should be clearly utilized to structure the results. Provide more clarity between the domains and how the themes/subthemes related to each domain. Tables 1 and 2 are not necessary as they do not add new information. This should help to reframe the results deductively within the CFIR domains.

** Thank you for highlighting this issue and providing an opportunity to clarify our process. In the first step of our analysis, we used the CFIR domains and constructs deductively in line-by-line coding. We also inductively developed lower-level nodes based on the data and integrated these into the CFIR domain framework. In the second step of our analysis, we met as a research team to review the codes, combining or collapsing codes where relevant, and discussed emerging ideas. In the third step of the analysis, we reviewed codes to identify patterns among first, second and third level codes, which were merged into categories. Because some factors were identified in more than one CFIR domain, in the final step of the analysis, overarching themes and subthemes were identified and reviewed by the research team. 

Using this process, the themes are not structured according to the five main CFIR domains. While many implementation studies that have used CFIR to present the facilitators and barriers based on the CFIR domains, others have used a similar approach to us, using CFIR to deductively code the data and then conduct a thematic analysis to identify facilitators and barriers (e.g., Norman, Å., Nyberg, G., Elinder, L. S., & Berlin, A. One size does not fit all-qualitative process evaluation of the Healthy School Start parental support programme to prevent overweight and obesity among children in disadvantaged areas in Sweden. BMC public health, 16, 37 (2016). https://doi.org/10.1186/s12889-016-2701-1; Kempen, T.G.H., Kälvemark, A., Sawires, M. et al. Facilitators and barriers for performing comprehensive medication reviews and follow-up by multiprofessional teams in older hospitalised patients. Eur J Clin Pharmacol 76, 775–784 (2020). https://doi.org/10.1007/s00228-020-02846-8).

We have made some revisions to the data analysis section of the manuscript to clarify the process used.

It has also been recommended to remove Tables 1 and 2 from the manuscript. We used these table to summarize the main themes and subthemes and highlight the alignment with the CFIR domains. To provide clarity on which CFIR domains and constructs contributed to each theme, we have added this information into the narrative text. Tables 1 and 2 have been removed.

In the manuscript that was originally submitted, we numbered the themes and used ‘theme’ and ‘subtheme’ labels to help readers link the information in the text with the information in Tables 1 and 2. As Tables 1 and 2 are no longer in the manuscript, we have removed the theme numbering and “theme” and “subtheme” labels.

10. Ensure that there is a more thorough narrative description of the results in relation to each domain/theme/subtheme.

** We have revised the manuscript to provide a more thorough narrative description of the findings. As described above, for each theme, we have also indicated which CFIR domain(s) contributed to the theme.

11. Include a clear connection between the narrative description of the findings and illustrative quotes.

** We have revised the manuscript to provide more of a connection.

12. Throughout the results, the term client is used. Clarify if you are referring to patients or caregivers, or both patients and caregivers.

**“Client” is used to refer to the older adults with multimorbidity and depressive symptoms who were transitioning from hospital to home. This has been clarified in the description of the CAST intervention under the Methods section.

In addition, we reviewed the manuscript to ensure the term has been used consistently. In two places, “and caregivers” was added when the issue being discussed was relevant to both clients and caregivers.

13. DISCUSSION: 

Ensure that the key results are highlighted and clearly connect to the results of this study. For example, discuss the finding related to challenges in engaging family physicians.

** The discussion has been revised to highlight the key findings. Challenges related to engaging family physicians is discussed.

14. Ensure that the key results are not just repeated but are discussed in relation to existing evidence. Ensure that the findings are interpreted in light of what the study adds to our current knowledge and/or how it confirms prior knowledge.

** The discussion has been revised to focus on key findings and interpret the findings based on existing literature. 

15. Include the implications of this research for policy and/or practice.

** Policy implications have been added to the end of the discussion.

16. EDITOR

While I understand that the focus of this study is the implementation of the intervention among professionals, stakeholders and the community at large, the missing perspective of the care recipients needs to be presented and justified already in the introduction. 

** We agree that the perspectives of the older adult clients and caregivers regarding the implementation of the intervention is an important perspective. In fact, post-intervention interviews with clients and caregivers were part of our evaluation plan. However, due to the complexity of the health conditions of many of the clients (and the impacts of this on caregivers), only a small number of clients and caregivers agreed to take part in an interview. As a result, we did not include these data in our analysis. We have highlighted this gap under Participants in the Methods section. We have also highlighted this as a limitation in the Strengths and Limitations section of the Discussion, although have revised this statement to provide more detail.

17. I would suggest to restructure the introduction to more focus on implementation issues instead of focusing on the care recipients. 

** Much of the original text in the Introduction related to older adults with multimorbidity and transitional care has been kept to provide context. However, a paragraph has been added that addresses the importance of examining the implementation of interventions.

18. I agree with the review to also present some descriptive statistics (age, nature and/or number of chronic conditions) on the target or actual population. Were there any inclusion or exclusion criteria to be enrolled in the intervention?

** The eligibility criteria to receive the CAST intervention are outlined in the Overview of the Pragmatic Trial (which has now been moved to the Methods section).

19. The methods section mentions that the intervention was tested in a randomized controlled trial, but I didn't see any details on how the care recipients were assigned to intervention or control arm, and any comparison between intervention and control group. I recommend to add an explanation that this study focuses on the intervention group and the experiences with the implementation of the intervention.

** The following sentence was added to the beginning of the Participants subsection within the Methods: “This study focuses on the implementation of CAST with older adults who received the intervention.”

20. I suggest some streamlining throughout the manuscript and particularly the discussion, as several details of the intervention and themes identified are repeated. 

** We have made revisions in an effort to streamline the manuscript.

21. The description of the intervention in the first section of the introduction and the study overview should be presented in the methods section.

** The description of the intervention and overview of the pragmatic trial have been moved to the Methods section.

** We have reviewed the style requirements and made adjustments where needed.

23. We note that the grant information you provided in the ‘Funding Information’ and ‘Financial Disclosure’ sections do not match. 

**Note – I intended to make this change but could not see where it should be done when I resubmitted.

24. We note that you have indicated that data from this study are available upon request. PLOS only allows data to be available upon request if there are legal or ethical restrictions on sharing data publicly. For more information on unacceptable data access restrictions, please see http://journals.plos.org/plosone/s/data-availability#loc-unacceptable-data-access-restrictions. 

** In accordance with the Hamilton Integrated Research Ethics Board (HiREB) for Hamilton Health Sciences and McMaster University’s Faculty of Health Sciences, data from this study cannot be shared publicly because of ethical restrictions involving potentially identifying information. The participant consent forms used in this study do not address open public access to the data. Data are available upon request from McMaster University, Faculty of Health Sciences, School of Nursing for researchers who meet the criteria for access to confidential data pending approval from the HiREB. For inquiries, please contact: Dr. Michael McGillion, Associate Professor and Assistant Dean, Research, School of Nursing, Faculty of Health Sciences, McMaster University, Email: mmcgill@mcmaster.ca; Phone: 905-525-9140 x 20275.

---

## [Editor Report · Decision Letter 1]

4 Jul 2022

Implementation of the Community Assets Supporting Transitions (CAST) transitional care intervention for older adults with multimorbidity and depressive symptoms: A qualitative descriptive study

PONE-D-21-07160R1

Dear Dr. McAiney,

We’re pleased to inform you that your manuscript has been judged scientifically suitable for publication and will be formally accepted for publication once it meets all outstanding technical requirements.

Kind regards,

Anja K Leist, Professor Dr.

Academic Editor

PLOS ONE

Additional Editor Comments (optional):

The reviewer and editor comments have been appropriately addressed. Additional revisions consolidating the results of the qualitative analyses and the more contextualised discussion have considerably improved the manuscript.
---

## [Editor Report · Acceptance letter]

27 Jul 2022

PONE-D-21-07160R1 

Implementation of the Community Assets Supporting Transitions (CAST) transitional care intervention for older adults with multimorbidity and depressive symptoms: A qualitative descriptive study 

Dear Dr. McAiney:

I'm pleased to inform you that your manuscript has been deemed suitable for publication in PLOS ONE. Congratulations! Your manuscript is now with our production department. 

Kind regards, 

on behalf of

Prof. Dr. Anja K Leist 

Academic Editor

PLOS ONE